# Effects of Drugs and Excipients on Hydration Status

**DOI:** 10.3390/nu11030669

**Published:** 2019-03-20

**Authors:** Ana M. Puga, Sara Lopez-Oliva, Carmen Trives, Teresa Partearroyo, Gregorio Varela-Moreiras

**Affiliations:** 1Department of Pharmaceutical and Health Sciences, Faculty of Pharmacy, CEU San Pablo University, 28668 Madrid, Spain; anamaria.pugagimenezazca@ceu.es (A.M.P.); sar.lopez.ce@ceindo.ceu.es (S.L.-O.); trilombar@ceu.es (C.T.); gvarela@ceu.es (G.V.-M.); 2Spanish Nutrition Foundation (FEN), 28010 Madrid, Spain

**Keywords:** dehydration, hydration status, water balance, elderly, drug interactions, excipients, polypharmacy, chronic treatment, adverse effects, pharmaceutical care

## Abstract

Despite being the most essential nutrient, water is commonly forgotten in the fields of pharmacy and nutrition. Hydration status is determined by water balance (the difference between water input and output). Hypohydration or negative water balance is affected by numerous factors, either internal (i.e., a lack of thirst sensation) or external (e.g., polypharmacy or chronic consumption of certain drugs). However, to date, research on the interaction between hydration status and drugs/excipients has been scarce. Drugs may trigger the appearance of hypohydration by means of the increase of water elimination through either diarrhea, urine or sweat; a decrease in thirst sensation or appetite; or the alteration of central thermoregulation. On the other hand, pharmaceutical excipients induce alterations in hydration status by decreasing the gastrointestinal transit time or increasing the gastrointestinal tract rate or intestinal permeability. In the present review, we evaluate studies that focus on the effects of drugs/excipients on hydration status. These studies support the aim of monitoring the hydration status in patients, mainly in those population segments with a higher risk, to avoid complications and associated pathologies, which are key axes in both pharmaceutical care and the field of nutrition.

## 1. Introduction

Among the different fields of physiology and nutrition, hydration does not always receive the attention it deserves. Water is essential for life and performs crucial functions in the human body, including nutrient transport through the circulatory system, tissue and joint lubrication, the maintenance of a stable body temperature, and as the medium that allows the chemical reactions of the organism to take place [1]. Regarding body composition, water represents around 80% of the body weight in a new-born. This percentage decreases with age, leading to values of 60% and 50% in adult men and women, respectively, although it is increased in vital organs, such as lungs or kidneys, to up to 80% [2,3]. Total body water comprises both intracellular and extracellular water and is highly dependent on the amount of lean mass and muscle [4,5]. Hydration status (HS) is a parameter indicative of total fluid composition. In this regard, water balance (WB), defined as the difference between water input and output, is the most common parameter used to evaluate HS. There are several mechanisms through which WB remains stable, such as increasing thirst sensation, compensation of water losses via vasopressin secretion, stimulation of the renin-angiotensin-aldosterone system, sympathetic activation or reduction of renal solute, and water excretion [6]. In an ideal situation, WB is null, i.e., water inputs are equal to outputs. However, under certain circumstances, an imbalance between water intakes and losses occurs, and WB acquires a negative sign, leading to hypohydration [2]. This may be classified as hypertonic, isotonic, or hypotonic. Hypertonic hypohydration appears when water losses exceed sodium losses, a condition that is mainly and frequently found in older people with infections and during periods of extremely hot weather. If water and sodium losses are similar, as in the case of vomiting or diarrhea, isotonic hypohydration occurs. Finally, in hypotonic hypohydration, sodium losses are greater than water losses. This kind of hypohydration is frequent in patients being treated with diuretics [7].

Our body can obtain water from three sources: (i) beverages, including water, juices, dairy products, and soft drinks; (ii) solid food, mainly fruits and vegetables; and (iii) in the metabolism of macronutrients [8]. Since the amount of water from metabolic processes is less than 300 mL, and the human body is unable to store water, it seems easy to understand that water must be obtained from the diet. According to the European Food Safety Authority (EFSA) [8], a suitable daily intake would be 2.5 and 2.0 L/day in men and women, respectively, regardless of age, including both water from beverages and that from solid food. It has been estimated that about 75–80% of water intake typically comes from liquids (i.e., around 1500 mL) and the remaining 20–25% from solid food [9,10]. While maximum water intake values do not exist, it is well known that very high intakes could lead to acute toxicity, owing to inadequate renal excretion [11]. On the other hand, water is eliminated via either urine, feces, sweating, or breathing and is highly dependent on internal and external factors, such as physical exercise, weather, or the coexistence of pathologies, for instance, infections, kidney diseases, and/or diarrhea [1,8,12]. Specifically, a recently published study showed that insensible water loss through skin and respiration increases with age, increasing the predisposition of the elderly to hypohydration [13].

Nowadays, there is no consensus about the most appropriate method to evaluate this pathology, particularly in the elderly. Several biomarkers have been proposed, such as blood and urinary indices, as well as the estimation of body water via bioelectrical impedance. Nevertheless, there are no markers which have been recognized as of a “gold standard”, because no single method appears to be ideal for the diagnosis of hypohydration [14,15,16,17,18]. Hypohydration consequences affect virtually every organ and system in the body and can cause urological disorders, constipation, circulatory disorders, anorexia, liver dysfunction, fatigue, low blood pressure, somnolence, increased body temperature, edema, and neurological disorders [19]. 

Population groups at higher risk of suffering from hypohydration are children, pregnant and lactating women, and the elderly [1,20,21,22]. In fact, it is considered that vulnerability to hypohydration is associated with aging [1]. In this last group, different factors coexist, increasing their hypohydration risk. Among them should be mentioned several physiological factors [23]. One of the most recognized characteristics of the elderly is their decrease of thirst sensation, owing to the diminished responsiveness of the osmorreceptors and the decrease of angiotensin I levels [12]. This situation worsens when neurological deterioration is also present, since hypothalamus and pituitary gland functions may lead to a reduction of thirst perception and incorrect regulation of liquids [24]. An altered renal function and the presence of chronic diseases are important conditions that may also affect HS [23]. Hypohydration in the elderly may also have different pathological causes, including fever, diarrhea, vomiting, coma, isolation, mental confusion, stroke, dementia, and cerebrovascular diseases [19,25]. In addition, dehydration can induce alterations in the renal function that can increase the risk of renally excreted drug toxicity (e.g., ribaroxaban, apixagan, digoxin, gliptins, oxycodone, morphine, or lithium salts) [26].

Not only internal, but also external factors, alter HS, including dietary compounds or drugs. For example, it is well known that alcohol is closely related to dehydration due to its diuretic effect and to the reduction in alertness, judgement, and perception of heat [27]. This effect may be sharpened by the concomitant consumption of certain drugs. It seems that a moderate consumption of low-grade alcoholic beverages, such as beer, do not compromise HS. However, an increase of either the dose or the alcohol content, may worsen hypohydration [28]. Similarly, caffeinated drinks as well its diuretic effects could also lead to hypohydration. However, this effect is not apparent at moderate intake and no significant differences in HS were noted between individuals with moderate caffeine intake and those who drank only water or had mild caffeine consumption. Thus, this diuretic effect is only expected in cases of high caffeine intake or habituation [29,30]. 

In addition, polypharmacy or the consumption of certain drugs, especially in cases of chronic treatments may also modify WB. Drug effects on HS must be considered, not only in the elderly, but also in professional or amateur athletes, since drugs could aggravate fluid loss during exercise or prevent the consumption of enough liquid to achieve the correct repositioning of fluids after exercise. Generally, it is considered that drugs induce hypohydration by means of three mechanisms: (i) Decreasing thirst sensation, which leads to a decrease in liquid intake, (ii) increasing liquid elimination via urine, feces or sweat, and (iii) altering central thermoregulation [31]. In addition, other drugs may affect HS due to sedation or increased confusion which may affect heat perception and judgement, hyperpyrexia associated with serotonin syndrome or hypersensibility reactions [32]. One of the most common mechanisms by which drugs increase water elimination is diarrhea [33], but it is essential to differentiate between acute diarrhea and chronic diarrhea. Acute diarrhea lasts less than four weeks and usually has an infectious origin. On the other hand, chronic diarrhea is classified into watery, inflammatory or fatty diarrheas, based on stool characteristics [34]. It has been estimated that more than 700 drugs can prompt diarrhea [33]. There are different mechanisms by which drugs induce this effect, including affecting certain regulatory pathways in the gastrointestinal tract, allergic reactions, causing injuries in the intestinal tissue, shortening the gastrointestinal transit time, protein-loss through enteropathy and/or malabsorption or poor digestion of lipids and carbohydrates, among other mechanisms [35]. Some of these mechanisms may also appear simultaneously. Likewise, certain drugs affect thermoregulation, owing to the alteration of the temperature, which is established by the hypothalamus. This disturbance may be due to the inadequate function of thermoreceptors, damage to heat production, altered cutaneous vasodilation or an impaired cholinergic system, affecting sweat production. 

To date, to the best of our knowledge, there is only one study that has focused on the evaluation of drug–water interactions. Specifically, this pilot study from our research group analyzed the influence of diuretics, corticoids, and metformin in HS in an elderly population. The results of this study demonstrated the predominant hypohydration state of the volunteers, which was affected not only by drug consumption, but also by their route of administration [36]. However, it is necessary to spread this study to deepen our knowledge and to achieve more consistent results. Likewise, to evaluate the overall effect of drugs in the HS, it is necessary to consider, not only the active ingredient itself, but also the pharmaceutical excipients. Concretely, this latter term refers to those inactive substances that facilitate their preparation for, and acceptance by, the patients as well as the suitable behavior of the dosage form, as a drug delivery system. Among pharmaceutical excipients, there are disintegrants, fillers/diluents, lubricants, emulsifiers, flavorants, colorants, preservatives, anti-adherents, etc. [37].

Therefore, the aim of this review is to increase the reader’s understanding of the effect of drugs and excipients on HS. For a better comprehension and follow-up, the drugs are presented according to the mechanism of HS alteration and the Anatomical Therapeutic Chemical (ATC) classification (Table 1). This well-known and internationally accepted method to classify drugs is based on the system or organ in which the drug exerts its pharmacological effect, its therapeutic indications and its chemical structure [38]. 

## 2. Alimentary Tract and Metabolism Drugs That May Affect Hydration Status

According to the ATC classification, group A includes drugs indicated for the treatment of pathologies of the digestive system and metabolism, including diabetes mellitus (DM). Metformin is the first-line drug treatment for type-2 DM. While metformin has an excellent safety profile, it may cause gastrointestinal disturbances, including diarrhea, nausea, and dyspepsia, in almost 30% of patients, after treatment initiation [40,42]. These effects may have important consequences in a patient’s life, including HS alteration, and may even cause the patient to give up treatment. Several mechanisms have been proposed to explain these effects. For example, the structural similarities between the metformin and selective agonists of serotonin receptors trigger alterations in the transport of this neurotransmitter or a direct serotonergic-like effect, associated with nausea, vomiting, and diarrhea [42]. This effect was confirmed by a comparison of duodenal biopsies from metformin-naive individuals and patients treated with this drug that showed a stimulation of serotonin release, caused by this drug [41]. To avoid adverse gastrointestinal effects, it is highly recommended to initiate metformin treatment at low doses and gradually improve tolerance. Moreover, the selection of controlled release formulations is also a good alternative since they rarely cause gastrointestinal issues [39]. On the other hand, metformin has been reported to increase luminal bile salt concentration, which would have an osmotic effect, hence leading to the development of diarrhea [43]. In addition, metformin decreases appetite, which may cause a decrease in water intake, mainly that from solid foods, such as fruits and vegetables [136,138]. Since the primary route of metformin elimination is the kidney, special care must be taken in patients with advanced stages of renal insufficiency and patients older than 80 years [140,141,142]. In patients with acute renal affectation, metformin may cause lactic acidosis due to drug accumulation [40]. Hypohydration is a risk factor for developing lactic acidosis, and this condition may worsen by concomitant treatment with angiotensin-converting enzyme inhibitors (ACEIs) or angiotensin II receptor blockers (ARBs) [136]. 

Sodium-glucose cotransporter 2 (SGLT2) inhibitors, also called gliflozin drugs, are modern oral antidiabetic agents, whose activity is based on the inhibition of glucose reabsorption in the renal proximal tubules and rise excretion of glucose into the urine, hence resulting in a lower blood glucose. Interestingly, they are able to lower blood glucose, without suppressing insulin secretion. In addition to their effect on glycemic parameters and weight loss, SGLT2 inhibitors are also able to reduce blood pressure, which is secondary to their loop diuretic action [112,113]. This effect increases the risk of dehydration, especially in elderly patients with a lower capacity to concentrate urine. This particular mechanism has been postulated to explain the cardiovascular protective effect of empaglifozin [112,114]. Combined treatment with SGLT2 inhibitors and metformin should be performed cautiously, with the gradual introduction of SGLT2 inhibitors and re-evaluation of diuretic use to avoid related complications, including hypohydration. 

Drugs for the treatment of digestive pathologies, such as antacids or laxatives, may also induce hypohydration. For example, magnesium-containing antacids (including trisilicate or magnesium hydroxide), bowel laxatives containing this ion, and laxatives with poorly absorbable solutes, such as lactulose, induce osmotic diarrhea. This appears when poorly-absorbable or low–molecular weight aqueous solutes are ingested and, due to their osmotic force, quickly pull water and ions into the intestinal lumen, resulting in diarrhea and, hence, hypohydration [44]. Likewise, misoprostol and bysacodyl (for the prevention and treatment of gastric and duodenal ulcers and constipation, respectively) or chenodeoxycholic acid (to dissolve gallstones) usually induce diarrhea due to different electrolyte disorders that cause net fluid disorders [33]. Specifically, misoprostol induced diarrhea in 14 to 40% of patients [45,46,47], and bysacodyl at standard and appropriate dosages caused abdominal pain and non-bloody diarrhea [48]. Moreover, diarrhea is the most common adverse effect of chenodeoxycholic acid, affecting up to 40–50% of patients treated with this drug. Ursodeoxycholic acid, an epimer of chenodeoxycholic acid also used for biliary lithiasis, seems to present a better tolerance and lower adverse effects [49]. 

Proton pump inhibitors (PPIs), prescribed for several acid-related conditions, have been associated with a high risk of developing microscopic colitis (MC), an inflammation of the colon that causes persistent watery non-bloody diarrhea, with a normal radiological and endoscopic mucosal appearance but microscopic abnormalities in the colonic mucosa. While initially lansoprazole was related to a higher likelihood of inducing MC [51], similar effects were found for other PPIs, such as omeprazole, esomeprazole, or pantoprazole [52]. Keszthelyi et al. [50] carried out a retrospective case-control study to evaluate PPI consumption in patients with documented MC, using as controls age- and gender-matched volunteers. Researchers found that drug consumption was significantly higher in patients with MC than in controls [50]. The simultaneous consumption of PPIs with other drugs that induce MC has been also examined, observing that the concomitant use of nonsteroidal anti-inflammatory drugs (NSAIDs) and PPIs was found to cause the highest risk of MC [53,54]. An analysis indicated that dysbiosis associated with acid suppression may contribute to the effect of PPIs and that the gastrointestinal effects of NSAIDs might exacerbate PPI-related MC [53]. However, the studies performed to date are limited due to their small sample size, so there is a need for large observational studies to examine the differential effect of specific PPIs and the potential existence of a dose-dependent association. In addition, PPIs could lead to bacterial overgrowth and enteric infections, including *Clostridium difficile* [55,56]. 

Olsalazine is an anti-inflammatory drug for the effective oral treatment of active ulcerative colitis and may be potentially beneficial for patients with Crohn’s disease. It is a prodrug, designed to deliver the active moiety, mesalazine, to the colon [143]. Despite the effectiveness of this delivery, it induces diarrhea by increasing water secretion in the intestinal lumen and accelerating the gastrointestinal tract. Different authors postulate that olsalazine-induced diarrhea may affect 12 to 25% of patients [33,34,143]. This is distinguishable from the one associated with inflammatory bowel disease, owing to its higher water content and the absence of blood. Moreover, it usually appears just after the initiation of therapy and shows a dose-dependent trend [143]. Other mechanisms postulated to explain olsalazine-induced diarrhea are the inhibition of the Na/K ATPase or the stimulation of bicarbonate and sodium chloride secretion in the ileum [33,34,57]. This latter effect was demonstrated for the first time by Kles et al. [57], who compared intestinal secretion caused by mesalazine and its prodrugs, including olsalazine. The results of this study showed that prodrugs containing azo linkages, such as olsalazine, increase secretion in ileum, hence leading to increased diarrhea, an effect not found to be associated with mesalazine itself. 

Finally, the use of drugs based on medicinal plants, either as prescription or self-medication, is considerably increasing in recent years [144]. While they are generally considered relatively innocuous for health, adverse reactions are documented in literature reviews [145], including adverse effects related to HS. For example, *Cassia acuitifolia* and *angustifolia* are widely used as laxatives, although their chronic consumption can lead to fluid and electrolyte disorders, among other complications [58,146]. Vanderperren et al. [58] reported a case of a woman, who consumed a herbal tea with *Cassia angustifolia* for a long time and developed liver failure and renal impairment as well as polyuria, which caused significant hypohydration. While it is an isolated case, greater follow-up should be made in relation to multiple herbal medicines, especially when used in combination, since they may increase the risk of allergies, adverse reactions or cross-reactivity with other chemical drugs and supplements. 

## 3. Cardiovascular System Drugs That May Affect Hydration Status

Diuretics are drugs that are widely employed to treat several conditions, such as hypertension, congestive heart failure, liver failure, nephrotic syndrome and chronic kidney disease. Owing to their intrinsic mechanism, diuretics increase water elimination via urine. Thus, hypohydration and several electrolyte disorders with significant clinical impact may occur [115]. Among the different electrolyte disorders, the ones related to sodium, i.e., hyponatremia or hypernatremia, are primarily WB disorders, caused by alterations in the antidiuretic hormone vasopressin. Liamis et al. [118] carried out a study on 5179 subjects, aged 55 years or more, to determine the prevalence and risk factors associated with electrolyte disorders, reporting a prevalence of 25.7% for at least one electrolyte disorder in subjects taking diuretics and the existence of an independent association between diuretics and different electrolyte disorders, depending on their mechanism of action. The results of this study showed that thiazide diuretics led to hyponatremia, hypokalaemia, and hypomagnesemia, whereas loop diuretics cause hypernatremia, hypokalaemia, and potassium-sparing diuretics, such as hyponatremia. All these electrolyte alterations have important consequences for HS. Therefore, special attention should be paid to patients treated with diuretics in extreme environmental conditions, such as heat waves, to avoid hypohydration consequences. Michenot et al. [116] evaluated the adverse drug reactions in patients older than 70 years, during the heat wave that took place in France in 2003, using data from the French Pharmacovigilance Database. According to their findings, metabolic adverse reactions, including hypohydration and hydroelectrolytic disorders, were the most frequent ones, diuretics and ACEIs being the main drugs responsible for those effects. Similar results were observed in other studies, performed in Australian hospitals [117]. 

ACEIs and ARBs are widely prescribed for different indications, such as hypertension or heart failure. Captopril, enalapril or lisinopril belong to the ACEIs family, whereas candesartan, losartan, valsartan, or olmesartan belong to the ARBs. Adverse reactions related to ACEIs and ARBs treatment include metabolite disorders and hypohydration, although these effects are thought to be dependent on the pharmacokinetic profile of these drugs [147]. Animal studies revealed that the renin-angiotensin system is essential in thirst perception, and the inhibition of this system causes a reduction in water and salt intake and urinary volume, after the chronic intracerebroventricular administration of losartan, and thus, hypohydration [122]. Despite the interest of these results, to the best of our knowledge no studies have yet been performed to evaluate these effects in humans. A second mechanism to explain ACEI-related hypohydration has also been postulated by Farraye et al. [59]. These authors stated for the first time, in 1988, that intestinal angioedema is associated with ACEI treatment, characterized by recurrent episodes of pain, abdominal distention, and watery diarrhea. Interestingly, it is critical to consider that many patients are treated with a combination of ACEIs and diuretics, either in a single drug delivery system or with two or more tablets, so that concomitant effects that alter WB may appear, hence increasing the risk of developing complications derived from hypohydration.

There is strong evidence that olmesartan (a drug belonging to ARBs family) is associated with enteropathy, which causes severe diarrhea in about 7% of patients [148], leading to hypohydration and even kidney failure, as stated in recently published reviews [60,61]. Rubio-Tapia et al. [62] carried out a study on 22 non-celiac volunteers in Minnesota and evaluated their unexplained chronic diarrhea and enteropathy, similar to celiac disease, while being treated with olmesartan. Intestinal biopsies showed villous atrophy, inflammation of the intestinal mucosa and marked subepithelial collagen deposition. After treatment withdrawal, volunteers experienced weight gain and the histological recovery of the duodenum. HS evaluation of these volunteers is especially crucial in those treated with a combination of olmesartan and hydrochlorothiazide, since a combination of two different mechanisms of hypohydration takes place. Moreover, it should be considered that not only olmesartan, but also other members of the ABRs family induce enteropathy and, hence, chronic diarrhea. For example, single cases of enteropathy and diarrhea have been reported in patients under treatment with irbesartan and valsartan [63,65]. Other studies revealed, however, that the incidence of enteropathy associated with treatment with ABRs is very low [64]. Therefore, further studies should be performed to elucidate not only this relation, but also the clinical relevance of this drug-nutrient interaction. 

Beta-blockers, such as propranolol or nebivolol, are a group of drugs used for the treatment of hypertension, angina pectoris, or heart failure and for the prevention of recurrent heart attacks. It has been hypothesized that these drugs may alter HS due to their effects on thermoregulation. The hypothalamus is the central integration center for thermoregulation, and the skin plays a substantive role in the thermoregulatory process. Under normal conditions, in response to increases or decreases of ambient or internal temperatures, blood flood is modified accordingly through vasodilatation or vasoconstriction, respectively [31,131]. However, non-thermal factors, mediated by reductions in the mean arterial or cardiac filling pressures, also affect cutaneous blood flow. Thus, since beta-blockers reduce mean arterial pressure, the relation between skin blood flow and body temperature may be altered, and thus, beta blockers might also reduce heat dissipation [132]. Furthermore, in healthy adults, the administration of propranolol during exercise under heat stress was found to decrease sweating [134]. Rivas et al. [135] carried out a study to evaluate the putative reduction in heat dissipation, mediated by propranolol, in burned patients, whose thermal homeostasis is disrupted due to the inability of burned skin to control heat loss. However, these researchers found that therapeutic propranolol does not negatively affect skin blood flow and further compromises temperature regulation in burned children. Therefore, more studies should be performed to analyze these controversial results, and special attention should be paid to the elderly treated with beta-blockers, owing to their greater vulnerability to thermoregulatory disorders.

Digoxin indications include mild-to-moderate heart failure, combined with a diuretic and an ACEI when possible. Digoxin has a narrow therapeutic margin, so monitoring it is highly recommended to avoid complications and poisonings. Digitalis toxicity may produce general systemic symptoms, specific cardiac arrhythmias and, even, conduction disturbances. Weakness, lethargy, anorexia, nausea, and vomiting are included among the common non-cardiac symptoms of digoxin toxicity, which may be attributed to drug combinations in the management of different pathologies involving this drug [66,67]. Moreover, different reviews have analyzed the relationship between digoxin and diarrhea [33,35]. According to Pentland and Pennington [70], the risk of diarrhea development is higher in the elderly group, since this drug is mainly excreted through the kidneys, and renal function tends to decrease with age. Digoxin-related diarrhea has been related to the inhibition of the Na/K ATPase, which is responsible for the water regulation and electrolyte transport in the enterocyte [33,35]. Increased attention should be paid to patients with heart failure and concomitant kidney disease, since they have an increased risk of developing digitalis toxicity, mainly if the drug is combined with diuretics that may cause electrolyte imbalance and modifications in HS [67]. Additionally, digoxin increases the risk, especially in the elderly, of developing other intestinal complications, such as ischaemic colitis, which is attributable to the decrease in the intestinal blood supply [69]. It is crucial to indicate that this disturbance could be a potential cause of abdominal pain, diarrhea, or bloody diarrhea, so as to identify any precipitating drugs, such as digoxin [68]. Lastly, it is important to note that digoxin also includes, among its adverse effects, appetite decrease. This, as stated before, may reduce fruits and vegetable consumption and, therefore, may affect HS [66,138].

Statins are the most widely prescribed group of cholesterol-lowering drugs that cause diarrhea in less than 5% of the patients [33]. However, these drugs render the organism more vulnerable to heat and its consequences, including hypohydration [71]. In addition, recent studies have analyzed the relationship between statins and MC, with contradictory results [51,53,72,73,74]. Particularly, simvastatin and fluvastatin have been classified as drugs with an intermediate likelihood of causing MC [74]. Bonderup et al. [72] carried out a case-control study on 3474 Danish volunteers, with MC diagnosis finding positive associations between statins consumption and this disease. Similar results were obtained by Fernandez-Bañares et al. [73], who suggested that statins and other drugs might be trigger the factors of colonic inflammation in predisposed hosts. However, in other patients, these drugs only worsen self-evolving MC. Nevertheless, a British study on 1211 cases with MC and 6041 controls was not able to associate statins consumption with MC [53]. Thus, there is a strong need to increase knowledge of the relationship between statins and MC, owing to its important consequences, not only in HS, but also in the overall health of the patients.

## 4. Genito-Urinary System Drugs That May Affect Hydration Status

According to the ATC (Anatomical Therapeutic Chemical) classification, Group G includes drugs used for the treatment of different pathologies associated with the genito-urinary system and sex hormones. Silodosin, usually prescribed for the treatment of benign prostatic hyperplasia, a non-cancerous increase in size of the prostate, also called prostate enlargement, have been related to alterations in HS [75]. A summary of the product characteristics by the European Medicines Agency include diarrhea as a frequent adverse effect, affecting around 10% of patients [149]. Magan-Martinez et al. [75] recently published a clinical case of a man that experimented severe diarrhea, after silodosin treatment, which was reversible after treatment suspension. However, Marks et al. [76] reported in a clinical trial on 661 patients, in which diarrhea appears only in 4% of the volunteers and is not described as a serious adverse effect. Despite the fact that the prevalence and magnitude of silodosin-induced diarrhea is controversial, additional attention should be given to patients treated with this drug, especially the older ones, to avoid complications derived from potential hypohydration.

## 5. Systemic Hormonal Preparations That May Affect Hydration Status

Corticoids are anti-inflammatory and immunosuppressive drugs that are administered orally, pulmonarily, nasally, and topically for the treatment of different pathologies, including intestinal inflammatory disease, asthma, rhinitis, systemic lupus erythematous, or to avoid the rejection of transplanted tissues and organs. Among other adverse effects, corticoids induce potent diuresis in animals [120] and in patients with heart failure [119]. An explanation for this effect is that the hemodynamic actions of corticoids alter autoregulation, increase the glomerular filtration rate, and promote the loss of sodium and potassium ions [119]. Furthermore, a recently published study by our research group [36] confirmed that patients under corticoids treatment have a higher potential to suffer from hypohydration. In fact, according to this study, the interactions between HS and corticoids are dependent on the route of administration of the drug (oral vs. pulmonary). These differences could be related to the different pharmacokinetic behaviors of the drug [36].

## 6. Anti-Infectives for Systemic Use That May Affect Hydration Status

It has been estimated that antimicrobials are responsible for approximately 25% of drug-induced diarrhea [34]. This iatrogenic affectation is especially relevant in elderly patients treated with broad-spectrum antibiotics [77]. The risk of developing antibiotic-associated diarrhea (AAD) varies greatly between studies, ranging from 2.1% to 80%, depending on different factors, such as the type of diarrhea and antibiotic or the population under study [84]. Usually, AAD appears during the first days of treatment and disappears at the end of the therapy [33]. However, cases of incubation times of eight weeks after discontinuation have been reported [150]. The mechanism underlying AAD is not fully understood. It has been hypothesized that it could be caused by a disturbance of the normal intestinal flora by specific enteric pathogens (i.e., *Clostridium difficile* or *Candida spp*) [77], or it could be the direct effect of the antibiotics in the intestinal mucosa [151]. Moreover, excessive overgrowth of opportunistic pathogens (such as *Staphylococcus aureus* and *Klebsiella oxytoca*) cause 15–39% of AADs [152]. A different mechanism that links the consumption of drugs and the appearance of diarrhea is the small intestine mucosa damage [35]. Hence, studies performed on patients under chronic treatment with the antimicrobial neomycin revealed different histological changes in their mucosa, including severe villous atrophy [79,153]. AAD mainly appears in the shape of benign diarrhea, which may induce hypohydration or worsen malnutrition in the elderly [154]. However, in severe cases, AAD may result in pseudomembranous colitis or a toxic megacolon, leading to increased fatality rates [152]. It is considered that almost every antibiotic family may lead to the appearance of this adverse effect, as demonstrated by the available studies that focus on the quinolone levofloxacin [78] or macrolides [82]. However, Gilles et al. [81] reviewed the appearance of any reported adverse event in epidemiological studies that evaluated the prescription of either amoxicillin or amoxicillin-clavulanic acid for common indications. According to their analysis, diarrhea appeared only when amoxicillin was consumed in combination with clavulanic acid. Thus, alterations in HS are not expected in patients treated only with this penicillin. Likewise, a study by Easton et al. [80] evaluated the use of amoxicillin/clavulanic acid for the management of pediatric patients with acute otitis media. These authors observed that, although gastrointestinal disturbances were the most frequently reported adverse events, differences in the frequency of appearance were dependent on the posology (drug dosage and time interval between successive administrations). Probiotics are routinely employed in clinical practice to treat this diarrhea, and different mechanisms have been proposed to understand their effect, including the activity in the intestinal lumen (competition or direct suppression of pathogenic microorganisms), interactions with the mucosal barrier (up-regulation of tight junctions or modulation of water and ion channels) or influence on the intestinal immune system [83,155]. In addition to ADD, antimicrobial, specifically quinolones, led to many signs of other adverse reactions, such as loss of appetite [137], that may have potential consequences in WB.

## 7. Antineoplastic and Immunomodulating Agents That May Affect Hydration Status

Mycophenolate mofetil (MM) is an immunomodulating agent, commonly used to avoid rejection after organ transplant. It has been estimated that approximately 50% of the cases of drug-induced diarrhea in solid organ recipients are caused by MM [85]. Specifically, Kamar et al. [86] reported four cases of iatrogenic diarrhea, subsequent to MM treatment. Despite the severity of the presented adverse effect, in those cases, diarrhea disappeared after MM removal. Histological analysis of the colonic mucosal biopsies of patients with MM-related diarrhea showed prominent crypt cell apoptosis, enterocyte cytologic atypia, increased neuroendocrine cells, and glandular architectural distortion, a pattern not seen in controls or the biopsies of patients that did not receive this drug. In addition, the observed pattern was significantly different from the one observed in idiopathic inflammatory or infectious colitis. This form of injury results, in the end, in a decrease of the mucosal protection that prevents diarrhea [87]. This same effect was observed in patients treated with other immunomodulatory drugs, such as azathioprine. This drug is used for rheumatoid arthritis and renal transplantation treatment. Among the off-label indications are lupus nephritis, Crohn’s disease, ulcerative colitis, and chronic refractory thrombocytopenic purpura. Ziegler et al. [88] detailed the case of a patient, who developed small-bowel villus atrophy and chronic diarrhea, subsequent to azathioprine treatment. This adverse effect led to micronutrient depletion, malnutrition, and important hypohydration, which required parenteral nutrition. Two weeks after azathioprine discontinuation, the diarrhea was completely resolved, indicating the reversibility of these adverse effects. 

Chemotherapy-induced diarrhea is a common effect of particular clinical importance, being reported in 50–80% of patients, depending on the chemotherapy regimen [156]. The exact mechanism by which diarrhea appears remains unclear, although risk factors contributing to direct toxic effects in the gastrointestinal tract are starting to be understood. In many cases, the physiological mechanism underlying chemotherapy-induced diarrhea is drug-dependent; however, relevant clinical research is scarce. Chemotherapeutic agents most frequently associated with diarrhea are fluorouracil (FU), a thymidylate synthase inhibitor, and irinotecan, a topoisomerase I inhibitor [157]. Despite the fact that studies to date have focused on FU-induced diarrhea in patients with gastrointestinal malignancies, patients with other tumors might also be affected. For instance, Schwab et al. [91] carried out a study on 683 patients with cancer treated with FU monotherapy and observed that 54% suffered from grade 3–4 diarrhea. Moreover, genetic background may also contribute to a drug-specific gastrointestinal toxic effect. Specifically, polymorphisms that regulate thymidylate synthase, methylenetetrahydrofolate reductase, and cytidine deaminase might contribute to the risk of diarrhea [90]. Capecitabin is the prodrug of FU that may effectively substitute parenteral FU for an oral treatment, but it can also induce diarrhea. Concretely, Iacovelli et al. [89] reviewed the incidence and relative risk of grade 3 and 4 diarrhea in patients treated with either FU or capecitabin. A higher risk of severe diarrhea was reported for capecitabin than for FU. These same authors observed that polychemotherapy increased the risk of this adverse effect. Thus, according to this metanalysis, a combination treatment of FU with irinotecan doubles the risk of this adverse effect, compared to that with FU treatment alone. Finally, the risk of FU-induced diarrhea is increased by a concomitant treatment with leucovorin. On the other hand, irinotecan is an inhibitor of the DNA polymerase used in the first- and second-line treatment of metastatic colorectal cancer. Myelosuppression and diarrhea are the most common side effects, reaching, in the case of the latter, incidences of 87% [158]. Irinotecan can induce either acute or delayed diarrhea. Acute diarrhea appears immediately after drug administration and is caused by the acute cholinergic effect of the drug. This adverse effect occurs along with other cholinergic symptoms, such as abdominal cramping, rhinitis, lacrimation and salivation. Delayed diarrhea occurs more than 24 h after drug administration, being non-cumulative and non-dependent on drug dose [156]. Several mechanisms have been proposed to explain irinotecan-induced diarrhea, including direct mucosal damage with water and electrolyte malabsorption, increased mucin secretion, and villous atrophy and crypt hypoplasia [92,93,156]. 

Recently introduced antineoplastic drugs, such as tyrosine kinase inhibitors, also lead to the occurrence of diarrhea. In fact, diarrhea is the second most common adverse event of this drug, after rash [157]. Interestingly, in this case, despite the fact that it is an adverse that may alter HS, the appearance of diarrhea could act as a predictable factor for tumor response [159]. Diarrhea usually appears just 2–3 days after the start of treatment, showing a dose-dependent trend. Diarrhea induced by tyrosine kinase inhibitors is thought to be related mainly to an excessive chloride secretion to the intestinal lumen, but it seems likely that more than one mechanism induces the diarrhea. Modified gut motility, colonic crypt injury, alterations to intestinal microflora, changed nutrient metabolism, absorption, and altered transport in the colon are considered as potential alternative mechanisms [157]. Specifically, idelalisib, an inhibitor of p110 delta phosphatidylinositol 3 kinase, used for malignancies in lymphoid tissues and bone marrow, has different adverse effects related to HS [95]. Diarrhea and colitis are included among the most frequent adverse effects reported in clinical trials [94,97]. Two different types of diarrhea have been observed in clinical trials with idelalisib: acute, mild, or moderate diarrhea that appears in the first eight weeks of treatment and delayed diarrhea that appears latter on and used to be resistant to both antimicrobial or antidiarrheal therapy [96]. Idelalisib diarrhea is usually watery, without cramping, blood, or mucus [98].

Finally, agents that interfere with biological regulatory molecules are increasingly used to induce tumor regression. For example, ipilimumab, a monoclonal antibody, approved for metastatic melanoma treatment, causes adverse gastrointestinal effects, including diarrhea or drug-induced enterocolitis, which have been reported to affect up to 44% of patients. Typically, these effects occur after treatment [99] and worsen after each subsequent dose [60]. Therefore, the cumulative data provided suggest that controlling the HS in patients with chemotherapy is crucial to avoid complications related to hypohydration that could that aggravate these patients’ conditions.

## 8. Musculoskeletal System Drugs That May Affect Hydration Status

NSAIDs (such as ibuprofen, acetylsalicylic acid, indomethacin, or diclofenac) are widely prescribed drugs, with anti-inflammatory, analgesic, and, some of them, anti-platelet effects. The prevalence of regular NSAIDs consumption in the United States population has grown in recent decades, reaching values of 10.4, 11.8, and 4.4% for aspirin, non-aspirin and multiple NSAIDs, respectively [160]. Data from other countries, such as Spain, revealed higher prevalences of consumption (11.2 and 29.6% for aspirin and non-aspirin NSAIDs, respectively) [161]. However, NSAIDs consumption varies, depending on different factors, such as ethnic group, age, or sex [160,161]. Likewise, self-medicated NSAIDs consumption is widespread [162], regardless of the age group, including the polymedicated elderly and other population groups, including athletes. NSAIDs are well known to increase the risk of gastroduodenal complications, such as peptic ulcer, bleeding, and perforations. Moreover, diarrhea is not infrequent, occurring in 3–9% of patients treated with several NSAIDs [163]. Likewise, in the last few years, interest in the effect of NSAIDs in the small bowel has increased. For example, Graham et al. [100] reported an endoscopically evident small-intestinal mucosal injury in 71% of NSAID users, compared to 10% in controls. While the pathogenesis of NSAID-induced enteropathy is not clearly understood, and symptomatology is non-specific, diarrhea is included among its clinical symptoms, and this could lead to alterations in the WB [101]. Moreover, NSAIDs have been also been related to the development of MC, characterized by aqueous diarrhea, as previously stated in this review. A study, performed by Verhaegh et al. [53], examined the association between the consumption of drugs, including NSAIDs, PPIs, and MC. Researchers observed that prolonged drug consumption, for 4 to 12 months, raised the risk of suffering from this pathology. The strongest association was found to have an average defined daily dose between 0.75–1.25, especially in the case of the concomitant consumption of NSAIDs and PPIs (5-fold increased risk). 

Colchicine is traditionally used for the prevention and treatment of the symptoms of gout or for the treatment of polyserositis. Colchicine may induce alterations of the WB, since it induces diarrhea in 80% of patients [105] by means of the inhibition of the Na/K ATPase in the enterocyte [35]. A different hypothesized mechanism to explain colchicine-induced diarrhea is the impairment of the mucosa of the small intestine. This disturbance was observed by Stemmermann and Hayashi [103] in patients chronically treated with this drug. Most of the studies, performed to evaluate the effect of colchicine in the gastrointestinal tract, focused on patients with chronic constipation. In this regard, Verne et al. [105] carried out a study to evaluate the effect of the drug on seven patients with chronic constipation. These researchers observed a significant increase in the number of spontaneous bowel movements as well as an acceleration of colonic transit. These same results were obtained in a randomized, double-blind, placebo-controlled crossover trial, performed on patients with chronic idiopathic constipation refractory to standard medical therapy [104]. However, no significant differences between conditions on ratings of nausea and bloating were observed. While these studies were performed on patients with chronic constipation, these effects should also be considered in patients with normal intestinal passage, since the potential acceleration of the colonic transit may affect their HS. In addition, long-term colchicine therapy has been associated with steatorrhea [102] and lactose malabsorption [164].

Auranofin is approved for the treatment of rheumatoid arthritis, although it is being investigated for other potential therapeutic applications, including cancer, neurodegenerative disorders, HIV/AIDS, parasitic infections and bacterial infections [165]. A study performed on patients with active rheumatoid arthritis revealed that around 74% of patients under long-term treatment with auranofin experienced at least one episode of diarrhea, which is a side effect that is especially significant in the early treatment. In fact, the monthly prevalence of diarrhea declined from 30–40% during the initial 6 months to about 10% for patients treated for 18–24 months [106]. The mechanism by which auranofin induces diarrhea is through the inhibition of the Na/K ATPase pump of the intestinal mucosa and the reduction of the absorption of bile acids [107]. 

Lastly, anthraquinones (9,10-dioxoanthracenes) constitute an important group of natural and synthetic compounds, with a wide range of therapeutic applications, including constipation, arthritis, multiple sclerosis, and cancer. Anthraquinones’ adverse effects include watery or factitious diarrhea production by means of the inflammation of the small intestine that results in an increase of water and electrolytes in the intestinal lumen, which, in the end, stimulate intestinal motility [34]. These water and electrolyte concentration alterations are thought to be related to the activation of the prostaglandin-cyclic AMP and the non-cyclic GMP pathways, which leads to the production of the platelet activating factor and to the inhibition of Na/K ATPase [166]. One example of an anthraquinone derivative used in clinical practice is the anti-inflammatory drug, diacerein [167]. Studies on this drug have mainly concentrated on its effects on joint-derived tissues/cells, which suggests a beneficial role in osteoarthritis treatment. Initially, it was proposed as an alternative to NSAIDs due to its different mechanism of action and fewer gastrointestinal adverse effects expected [168,169]. However, it was observed that diacerein efficacy was moderate, and it causes side effects in the lower digestive tract, including diarrhea. It has been estimated that around 36% of patients treated with this drug experienced even severe diarrhea as a side effect, occurring mainly during the first two weeks after treatment initiation [169]. The benefit/risk ratio of this drug has been questioned in recent years due to its important side effects, including diarrhea. In fact, there are discrepancies in this collateral laxative effect [169,170]. In any case, in 2013, the European Medicines Agency Pharmacovigilance Risk Assessment Committee recommended the suspension of the marketing authorization of diacerein across Europe, because its harms (particularly the risk of severe diarrhea and potentially harmful effects on the liver) outweighed its benefits [171]. 

## 9. Nervous System Drugs That May Affect Hydration Status

The treatment of Parkinson’s disease, a chronic and progressive neurodegenerative disease, with important morbidity and several social and economic consequences, was initially carried out with anticholinergics. However, currently, levodopa (L-DOPA), a metabolic precursor of dopamine, is the gold standard therapy. L-DOPA is usually combined with DOPA decarboxylase inhibitors, such as carbidopa or benserazide and, more recently, with catechol-O-methyltransferase inhibitors, such as tocalpone or entacapone. Triple therapy is also often used for Parkinson treatment. In any case, either double or triple therapy with enzymatic inhibitors result in an increase of L-DOPA half-life and enhanced drug plasmatic concentrations [172]. While in general, this therapy is well tolerated, diarrhea is reported among the adverse effects. In fact, significant differences in the appearance of this adverse effect were found in treated patients, compared to controls (20% versus 7%, respectively) [109]. Moreover, patients treated with the triple therapy experienced diarrhea in a higher percentage than those treated with L-DOPA and carbidopa or benserazide alone. Hence, these effects are thought to be related to catechol-O-methyltransferase inhibitors (tolcapone or entacapone). A review by Kaakkola [173] revealed that tolcapone causes severe diarrhea more frequently than entacapone. This author described entacapone diarrhea as explosive [174]. According to Larsen et al. [108], 14% of patients treated with this therapy discontinued treatment owing to the adverse effects, diarrhea being one of the most cited [172]. Furthermore, L-DOPA has been reported to reduce appetite [138], with subsequent consequences for water intake. In addition to the anti-Parkinsonian drugs previously mentioned, others, such as trihexyphenidyl, tropatepine, and biperidene, may negatively affect the production of sweat, hence disturbing thermoregulation. Lastly, dopamine D1- or D2-receptor-agonists, used to treat movement disorders, reduce thirst sensation [124]. Since Parkinson patients receive not only these drugs, but also others, including anti-psychotics or antidepressants, and frequently exhibit impaired mobility, this population group is especially vulnerable to suffering from hypohydration consequences. 

Several drugs from group N of the ATC (Anatomical Therapeutic Chemical) classification have been reported to induce the so-called syndrome of inappropriate antidiuretic hormone secretion (SIADH), characterized by a sustained release of the antidiuretic hormone from the posterior pituitary. This leads to a reduction in the patient’s ability to excrete water, hence resulting in hyponatremia, hypoosmolality, and thus, a decrease of thirst sensations and alterations in the HS [71,175]. Specifically, selective serotonin reuptake inhibitors (SSRIs), the most widely prescribed antidepressants, cause SIADH in between 0.5 to 32% of patients, with a higher prevalence in the elderly underweight patients, which are also under treatment with diuretics [176]. Letmaier et al. [123] monitored 263,864 psychiatric patients, analyzing their plasma sodium levels, and found hyponatremia in patients under treatment with SSRIs, carbamazepine and oxcarbazepine, which reported the highest hyponatremia rate. This study also revealed that combinations of these drugs with others that cause hyponatremia (such as diuretics or ACEIs) significantly increase the risk of developing this disturbance. Moreover, studies have tried to elucidate the relationship between SSRIs and MC, characterized by chronic or intermittent watery diarrhea that may lead to hypohydration, as previously stated [53,73,74]. In all cases, positive associations between drug consumption and MC were found. However, differences have been stated between moieties. Thus, sertraline was classified as a drug with a high likelihood of causing MC, whereas paroxetine was stated as a drug with an intermediate probability of inducing this injury [74]. Moreover, it has been reported that fluoxetine may reduce appetite, with subsequent consequences for HS [138,139]. Likewise, other antidepressants, including citalopram, clomipramine, duloxetine, venlafaxine, and mirtazapine, are responsible for causing SIADH, thus reducing thirst sensation [125].

Neuroleptic malignant syndrome is an idiosyncratic reaction to antipsychotic drugs, characterized by fever, altered mental status, muscle rigidity, and autonomic dysfunction that may even threaten a patient’s life. Hypohydration and malnutrition may increase the risk of this syndrome [177]. Moreover, neuroleptics and atypical antipsychotics are well known drugs that are prone to clinically relevant thermoregulation disturbances as well as a decrease of thirst sensation. For example, clozapine has been reported to have a thirst-blocking effect [126,127]. However, the effect on thirst sensation of other antipsychotic drugs, such as risperidone or olanzapine, is controversial [178,179,180]. Regarding thermoregulation disorders, psychiatric patients on antipsychotic drugs deserve special attention, especially during heatwaves, since both thirst sensations and thermoregulation are affected [133]. 

While electrolyte disorders are relatively common in the elderly, some drugs have been reported to induce them, with an associated increase in morbidity and mortality, including HS alterations. A study carried out by Liamis et al. [118] on 5173 non-hospitalized subjects aged 55 years or more evaluated the prevalence and risk factors of electrolyte disorders. In this study, researchers observed that patients under treatment with benzodiazepines, well-known anxiolytic drugs prescribed to reduce anxiety and insomnia, suffered from hyponatremia. In addition, researchers observed synergistic effects between benzodiazepines and diuretics and significant differences between sodium serum levels in patients treated with both thiazides and benzodiazepines, compared to those treated with one or none of these drugs. Regarding the biochemical mechanisms by which benzodiazepines induce hyponatremia, it seems that the gamma-aminobutyric acid (GABA)-ergic activity of benzodiazepines may be involved in vasopressin regulation [181]. Moreover, benzodiazepines are usually prescribed for the treatment of sleep disorders, which are also related to hyponatremia. An explanation for this relation is that lack of sleep decreases cortisol levels that increase vasopressin levels and, in the end, reduce sodium serum levels [182]. Likewise, benzodiazepines have been also related to thermoregulation disturbances, as demonstrated by Martin-Latry et al. [133], who indicated that anxiolytic consumption raises the risk of heat-related pathologies. 

Finally, other drugs, used for the treatment of nervous system pathologies, also affect HS by different mechanisms. Specifically, lithium salts are associated with a higher incidence of polyuria, diarrhea, enuresis, and polydipsia than other antipsychotic drugs, such as valproate [110] and reduced appetite [138].

## 10. Respiratory System Drugs That May Affect Hydration Status

Drugs indicated for the treatment of respiratory system pathologies may also trigger alterations in WB. Theophylline is a xanthine alkaloid, used for the treatment of respiratory diseases, such as asthma, chronic bronchitis, or emphysema, since it relaxes bronchial smooth muscle. This drug induces diarrhea by increasing cyclic AMP, opening chloride channels and, thus, increasing secretion to intestinal lumen [111] and altering WB. Moreover, despite the fact that SIADH and the subsequent hyponatremia are not usually included in the list of theophylline’s adverse effects, few studies have reported cases of this adverse reaction after treatment initiation [121,128,129,130]. Likewise, theophylline raises urine production, similar to thiazides, by means of the inhibition of solute reabsorption in both the proximal nephron and the diluting segment [121].

Finally, anticholinergic drugs are used to treat asthma, chronic bronchitis, and chronic obstructive pulmonary disease, etc. These drugs may alter thermoregulation by means of the reduction of heat elimination [133].

## 11. Excipients That May Affect Hydration Status

In the last few years, there has been an increasing interest in the impact of pharmaceutical formulations in human gastrointestinal transit [183]. Excipients have historically been considered inert, since they neither exert any therapeutic or biological action by themselves, nor do they modify the biological action of the drug. However, nowadays, it is considered that they may influence absorption properties and drug bioavailability, depending on the dosage form [37]. In this sense, there are excipients that induce osmotic diarrhea, such as sorbitol [33,158,184,185], mannitol [33,158,186], xylitol [33,158], fructose [33,158], phosphates [34,158], polyethylene glycol [187], and magnesium salts [33,35,158] (Table 2). 

On the other hand, the existence of pharmaceutical excipients that modify gastrointestinal absorption, the so-called critical pharmaceutical excipients or absorption-modifying excipients, owing to their capacity to alter the integrity of the intestinal epithelial cell membrane, should be highlighted [188,189,190]. Specifically, recently published studies demonstrated that some of these excipients, i.e., chitosan and sodium lauryl sulphate, increase intestinal permeability and subsequent absorption and bioavailability [188,189]. In this connection, Labrasol^®^ (a lipoidal excipient) and its salts, combined with medium-chain fatty acid at low concentrations, also raises intestinal permeability due to mild mucosal perturbation [190]. 

Furthermore, it is important to consider not only the excipients, but also the pharmaceutical dosage forms due to their influence in the gastrointestinal transit, particularly for oral modified-release formulations [191]. Therefore, it is crucial to consider these excipients and dosage forms and their potential consequences for the HS of vulnerable patients, such us pediatric and geriatric, and those critically ill.

## 12. Strengths and Limitations

Despite the fact that they play a key role in the maintenance of a suitable HS for a healthy ageing, drug/excipients-water balance interactions have not been deeply researched to date. It is important to notice that hypohydration and therapy failure may have a synergic effect, raising the morbi-mortality of the treated population, with a subsequent impact on the health system economy.

On the contrary, one of the limitations of this review is its narrative design. The lack of studies published to date on this topic and the high number of drugs that potentially alter HS prevented the systematic design of this review. Likewise, the few available studies on this matter limited the quantification of the impact of drug effects as well as estimation of the prevalence of adverse effects in certain cases.

## 13. Conclusions

The available results suggest that drugs and excipients play an important role in the maintenance of hydration status. Reported evidence shows that a careful review of the patient’s medication is crucial for the diagnosis of drug-induced hypohydration. Specifically, there are increasing data supporting the view that certain drugs, frequently consumed by the elderly (including antihypertensives, hypolipemic, hypoglycemics, or drugs for acid- or nervous-related disorders) may induce water balance alterations. Likewise, despite the fact that excipients are considered historically inert, the presented data confirmed that they may have a synergic effect on hydration status when combined with certain drugs. Related to this connection, it is also necessary to consider the pharmaceutical dosage form, owing to its influence in the gastrointestinal transit. Therefore, binomial hydration/pharmacotherapy may have direct consequences for patients’ health, since pharmacological treatment may alter hydration status, and this may be determinant for the treatment’s efficacy. Hence, hypohydration and therapy failure may have an impact on patient morbidity and mortality as well as on health systems. Considering all the above, medical doctors, pharmacists, and dieticians must work in a coordinated way to establish specific nutritional and hydration guidelines for patients under treatment with certain drugs.

## Figures and Tables

**Table 1 nutrients-11-00669-t001:** Summary of mechanisms and drugs that induce alterations in the hydration status.

Mechanism of Hydration Status Alteration	Anatomical Therapeutic Chemical Group	Drug	References
Diarrhea	Alimentary tract and metabolism	Metformin	[39,40,41,42,43]
		Magnesium-containing antacids and laxatives	[44]
		Lactulose	[44]
		Misoprostol	[45,46,47]
		Bysacodyl	[48]
		Chenodeoxycholic acid	[33,49]
		Proton pump inhibitors	[50,51,52,53,54,55,56]
		Olsalazine	[57]
		*Cassia acuitifolia and angustifolia*	[58]
	Cardiovascular system	Enalapril	[59]
		Olmesartan	[60,61,62,63,64]
		Irbesartan	[63,65]
		Valsartan	[63,65]
		Digoxin	[33,66,67,68,69,70]
		Statins	[51,53,71,72,73,74]
	Genito-urinary system and sex hormones	Silodosin	[75,76]
	Anti-infectives for systemic use	Antibiotics	[77,78,79,80,81,82,83,84]
	Antineoplastic and immunomodulating agents	Mycophenolate mofetil	[85,86,87]
		Azathioprine	[88]
		Fluorouracil	[89,90,91]
		Capecitabine	[89]
		Irinotecan	[92,93]
		Idelalisib	[94,95,96,97,98]
		Ipilimumab	[99]
	Musculoskeletal system	Nonsteroidal anti-inflammatory	[53,100,101]
		Colchicine	[102,103,104,105]
		Auranofin	[106,107]
	Nervous system	Levodopa combined with carbidopa or benserazide and/or tocalpone or entacapone	[108,109]
		Selective serotonin reuptake inhibitors	[53,73,74]
		Lithium salts	[110]
	Respiratory system	Theophylline	[111]
Increase of urine volume	Alimentary tract and metabolism	Sodium-glucose cotransporter 2 inhibitors (empaglifozin)	[112,113,114]
	Cardiovascular system	Diuretics	[115,116,117,118]
	Systemic hormonal preparations	Corticoids	[36,119,120]
	Nervous system	Lithium	[110]
	Respiratory system	Theophylline	[121]
Decrease of thirst sensation	Cardiovascular system	Angiotensin-converting enzyme inhibitors	[122]
	Nervous system	Selective serotonin reuptake inhibitors	[123]
		Dopamine D1- or D2-receptor agonists	[124]
		Citalopram, clomipramine, duloxetine, venlafaxine and mirtazapine	[125]
		Clozapine	[126,127]
		Benzodiazepines	[118]
	Respiratory system	Theophylline	[121,128,129,130]
Central thermoregulation affectation	Cardiovascular system	Beta-blockers (propranolol and nebivolol)	[31,131,132]
	Nervous system	Antipsychotics	[133]
		Anxiolytics	[133]
	Respiratory system	Anticholinergic	[133]
Increase of sweat production	Cardiovascular system	Beta-blockers (propranolol)	[134,135]
	Nervous system	Trihexyphenidyl, tropatepine and biperidene	[71]
Decrease of appetite	Alimentary tract and metabolism	Metformin	[136]
	Cardiovascular system	Digoxin	[66]
	Anti-infectives for systemic use	Quinolones	[137]
	Nervous system	Fluoxetine	[138,139]
		Lithium salts	[138]

**Table 2 nutrients-11-00669-t002:** Summary of mechanisms and excipients that induce alterations in the hydration status.

Mechanism of Hydration Status Alteration	Excipients	References
Osmotic diarrhea	Sorbitol	[33,158,184,185]
	Mannitol	[33,158,186]
	Xylitol	[33,158]
	Fructose	[33,158]
	Phosphates	[34,158]
	Polyethylene glycol	[158,183,187,192,193,194]
	Magnesium salts	[33,35,158]
Increase intestinal permeability	Chitosan	[188,189]
	Sodium lauryl sulphate	[188,189]
	Labrasol^®^ and its salts combined with medium-chain fatty acid	[190]

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
