# Peer review of "Effects of Drugs and Excipients on Hydration Status"

_nutrients, 2019, doi:10.3390/nu11030669_

Reviewer 1 Report

This is an interesting topic on and several reference show both positive and negative outcomes. Dehydration in the elderly is common and is not given sufficient attention in research.

The topic is appropriate and needful as a lot of older adults are on many medications, understanding the impact of various medication on hydration status is essential. The review addresses a rather unexplored area in hydration.

I particularly like Table 1 which summarises the mechanism and drugs that induce alterations in the hydration status. it is well referenced. Well done!

Where acronyms are used, the full meaning should be repeated after a few paragraph to make it easier for readers to follow the work.

for example, ATC was introduced and described in line 135. It was used again in line 336 and I had to go back looking for the meaning.

line 335: Using simple words/expressions like ' loss of sodium ion' instead of 'natriuresis' or 'loss of potassium ion' instead of 'kaliuresis' will be helpful.

I however will recommend that extensive English language changes. The review is packed with information that is needful, however the content is not too easy to read in its current form. There are lots of technical terms that are not simplified, lots of acronyms that may need to be changed slightly.

Overall, a good piece of work and highly commendable. 

Some examples of changes that will make it better:

Abstract

line 15: remove  the word 'is' and replace with 'being'

Introduction

line 38: remove the word 'arises' and replace with 'increased'

line 52: remove the word 'appears' and replace with 'occurs'

line 88: the word 'insolation' needs to be replaced with another. the meaning with regards to hydration/dehydration is not clear.

line 106, 118: the word 'affectation' is not clear in the context of hydration. can it be replaced by another word, please.

line 133: I believe the authors' aim is to increase our understanding of the effect fo drugs and excipient on the HS. currently it reads that ' the aim of this review is to deepen the effect of drugs and excipients on the HS'.

line 377: I believe the authors meant 'under nutrition/malnutrition' not 'denutrition'

line 389: not sure what 'posology' means

Conclusion

line 652: replace 'are' with 'being' and 'date' with 'data'

line 653: insert 'it' before 'that may have....'

 a general read through again will be great!

Author Response

This is an interesting topic on and several reference show both positive and negative outcomes. Dehydration in the elderly is common and is not given sufficient attention in research.

The topic is appropriate and needful as a lot of older adults are on many medications, understanding the impact of various medication on hydration status is essential. The review addresses a rather unexplored area in hydration.

I particularly like Table 1 which summarises the mechanism and drugs that induce alterations in the hydration status. It is well referenced. Well done!

We sincerely appreciate the comments of the reviewer.

 Where acronyms are used, the full meaning should be repeated after a few paragraph to make it easier for readers to follow the work. for example, ATC was introduced and described in line 135. It was used again in line 336 and I had to go back looking for the meaning.

The ATC (Anatomical Therapeutic Chemical) classification is a well-known and internationally accepted method to classify drugs based on the system or organ in which the drug exerts its pharmacological effect, its therapeutic indications and its chemical structure. However, according to your kind proposal we included the meaning of this acronym in different parts of the review (lines 383 and 626).

 line 335: Using simple words/expressions like ' loss of sodium ion' instead of 'natriuresis' or 'loss of potassium ion' instead of 'kaliuresis' will be helpful.

According to the reviewer´s suggestion, expressions were included in the text (line 406). 

I however will recommend that extensive English language changes. The review is packed with information that is needful, however the content is not too easy to read in its current form. There are lots of technical terms that are not simplified, lots of acronyms that may need to be changed slightly. Overall, a good piece of work and highly commendable.

Thank you for your suggestions to improve the quality of our manuscript.

 Some examples of changes that will make it better:

Abstract

line 15: remove  the word 'is' and replace with 'being'

The suggested change has been included in the new version of the manuscript.

 Introduction

line 38: remove the word 'arises' and replace with 'increased'

The word has been substituted in the text. Thank you. 

 line 52: remove the word 'appears' and replace with 'occurs'

The text has been modified following your kind suggestion.

 line 88: the word 'insolation' needs to be replaced with another. the meaning with regards to hydration/dehydration is not clear.

This mistake was corrected in the new version of the manuscript. Thank you.

 line 106, 118: the word 'affectation' is not clear in the context of hydration. can it be replaced by another word, please.

According to the reviewer´s suggestion, the word “affectation” was substituted in the new version of the review. 

 line 133: I believe the authors' aim is to increase our understanding of the effect of drugs and excipient on the HS. currently it reads that ' the aim of this review is to deepen the effect of drugs and excipients on the HS'.

The corresponding sentence has been rewritten in the new version of the manuscript (lines 159-160).

 line 377: I believe the authors meant 'under nutrition/malnutrition' not 'denutrition'

The mistake kindly noticed by the reviewer has been corrected in the new version of the manuscript.

 line 389: not sure what 'posology' means

Attending to your proposal, a statement explaining the meaning of posology is now included in the manuscript. 

Conclusion

line 652: replace 'are' with 'being' and 'date' with 'data'

Replacements suggested by the reviewer have been included in the new version of the manuscript.

 line 653: insert 'it' before 'that may have....'

The text has been modified, thank you.

 a general read through again will be great!

Reviewer 2 Report

It is an excellent review, I only suggest adding a section or at least a paragraph regarding the drugs consumed by professional or amateur athletes, which could aggravate the fluid loss during the exercise or prevent the consumption of enough liquid to achieve the correct repositioning of fluids after exercise.

Author Response

It is an excellent review, I only suggest adding a section or at least a paragraph regarding the drugs consumed by professional or amateur athletes, which could aggravate the fluid loss during the exercise or prevent the consumption of enough liquid to achieve the correct repositioning of fluids after exercise.

 We are very grateful to the reviewer for his/her comments of our manuscript. Moreover, according to his/her suggestion we included a paragraph in the Introduction section (lines 126-129) of the manuscript, indicating the importance of considering the effect the reviewed drugs in the hydration status of athletes. This fact was also mentioned in section 8 of the review owing to the important role of non-steroidal anti-inflammatory drugs self-prescription (lines 535-536).

Reviewer 3 Report

The article is addressing an important topic and is thoroughly referenced.

There are some areas requiring addressing as detailed below:

 English needs improving e.g. very first line of paper in the Abstract could be worded better:

 ‘Despite water is the most essential nutrient, so far it is highly forgotten in both pharmacy and nutrition fields’

The terms ‘dehydration’ and ‘hypohydration’ differ and in the manuscript, hypohydration is a more correct term to use. Dehydration is the process, hypohydration is the state.

Page 2

One can be in water balance and not in euhyudration e.g. someone in a hypohydrated state e.g. 3% body mass loss and still have fluid intake matching fluid loss thus maintaining a hypohydrated state.

Line 61

Not ‘must come’ but rather ‘typically comes’

Line 69

It is highly debated – provide a reference supporting the statement that 2% threshold is universally accepted.

Line 78 citation needed

Line 85-86 Rewrite the sentence – grammatical issues.

Line 92 – not ‘the HS’, merely ‘HS’

Line 93,94 citation needed

Line 98 Replace ‘lead as well’ with ‘also lead to’

Line 111 – elaborate on the differences between the forms of diarrhoea

Lines 118, 318, 492 affectation – not the correct word

Line 133 deepen – incorrect word

Table 1 is well put together but Muscle-skeletal system – should be musculoskeletal

Line 154 have – has

Line 158 Metformin elimination

Line 161 ‘This effect’ – be more specific/less ambiguous – which effect are you referring to? How does it link to negative HS?

Line 182 confusing sentence – reword.

Lines 183, 210 – Good – specifies the prevalence of the impact of the drug, whereas many of the other sections of the review fail to identify the prevalence of the impact of the drug e.g on diarrhoea. The risks daintified throughout the review are rarely quantified.

Line 196-7 need citation

Line 268 ABR has not been defined

Line 280 – that vs than

Line 287 The hypothalamus

Line 301 thermoregulation vs thermoregulatory

Line 320 – how important is this reduction in appetite though in terms of HS?

Line 333 grammar

Line 345 being controversial. ‘Additional’ rather than ‘extraordinary’.

Line 356 – citation needed – which study?

Line 361 citation needed

Line 371 What are the other contributors to AAD according to this/other studies

Line 373 Links

Line 378 Increased fatality rates

Line 442 Occurs

Line 467 ‘effects occur’

Line 470 ‘is crucial to’. What are the complications referred to here?

Line 474 – how prevalent is NSAIDs use?

Line 492 chronically

Line 509 – Was this chronic or acute diarrhoea? You mention the importance of the distinction in the introduction, but then do not really address this issue in the remainder of the review.

Line 558 Specify these consequences

Line 564 SIADH

Line 607 – what is a ‘high incidence’?

Line 612 ‘since it relaxes’ ‘These drugs’

Line 617 ‘Raises’ …..’similarly to’ not ‘with’

Line 631 ‘highlighted that’

Line 652-3 grammatical issues

Line 654 – sentence is a bit confusing – rewording required e.g. ‘contemplate’

Line 659 coordinated

Overall response

This is an interesting review of a little explored topic. It could be improved by identifying the limitations of the review e.g. narrative review as opposed to a systematic review. Also the quantification of the impact of these e.g. on balance for each drug/drug class - are these effects minor, moderate, major magnitude? Similarly the prevalence of these side effects is only occasionally addressed - could be more consistently addressed where the data supports.

Author Response

The article is addressing an important topic and is thoroughly referenced. There are some areas requiring addressing as detailed below.

English needs improving e.g. very first line of paper in the Abstract could be worded better:

 ‘Despite water is the most essential nutrient, so far it is highly forgotten in both pharmacy and nutrition fields’.

We sincerely appreciate the comments of the reviewer as well as his/her suggestions to improve our manuscript. According to your suggestion, we have corrected grammatical errors in this sentence.

The terms ‘dehydration’ and ‘hypohydration’ differ and in the manuscript, hypohydration is a more correct term to use. Dehydration is the process, hypohydration is the state.

Thank you for your suggestion. We have reviewed the manuscript and replaced the incorrect terms.

 Page 2 One can be in water balance and not in euhyudration e.g. someone in a hypohydrated state e.g. 3% body mass loss and still have fluid intake matching fluid loss thus maintaining a hypohydrated state.

Thank you very much for your comment. According to it, we have removed this mistake from the manuscript.

 Line 61 Not ‘must come’ but rather ‘typically comes’

According to the reviewer´s suggestion, this mistake was corrected in the new version of the manuscript.

 Line 69 It is highly debated – provide a reference supporting the statement that 2% threshold is universally accepted.

Thank you for your comment. We have removed this statement from the manuscript. 

 Line 78 citation needed

We have included specific references.

 Line 85-86 Rewrite the sentence – grammatical issues.

According to your recommendation, we have rewritten this sentence.

 Line 92 – not ‘the HS’, merely ‘HS’

We have corrected this mistake. Thank you.

 Line 93,94 citation needed

The reference was included in the new version of the manuscript.

 Line 98 Replace ‘lead as well’ with ‘also lead to’

Expressions were replaced in the new version.

Line 111 – elaborate on the differences between the forms of diarrhoea

According to your recommendation, we have elaborated the differences between acute and chronic diarrhoea.

Lines 118, 318, 492 affectation – not the correct word

According to the kind suggestion of the reviewer, we have substituted this word in the new version of our manuscript.

Line 133 deepen – incorrect word

Thank you for your observation. We have rewritten this sentence.

Table 1 is well put together but Muscle-skeletal system – should be musculoskeletal

This mistake was corrected in the new version of the manuscript both in the table and in the title of section 8 (line 529). Once again, thank you very much for the observation.

Line 154 have – has

The mistake kindly noticed by the reviewer has been corrected in the new version of the manuscript.

Line 158 Metformin elimination

We thank the reviewer for his/her comments on this point.

Line 161 ‘This effect’ – be more specific/less ambiguous – which effect are you referring to? How does it link to negative HS?

According to the reviewer´s indication, we have specified in the new version of the manuscript the effect we want to mention (lactic acidosis). Lactic acidosis is linked to negative HS since hypohydration is a risk factor for developing this potentially fatal condition.

Line 182 confusing sentence – reword.

Thank you very much for your suggestion. We have simplified this sentence.

Lines 183, 210 – Good – specifies the prevalence of the impact of the drug, whereas many of the other sections of the review fail to identify the prevalence of the impact of the drug e.g on diarrhoea. The risks daintified throughout the review are rarely quantified.

Effectively, as you mention, in several cases we included the prevalence of the impact of the drugs in adverse effects. However, in other cases, we did not include prevalence data since they are not available. In any case, we have included some prevalences not included in the previous version of the manuscript.

Line 196-7 need citation

Thank you for your observation. We have included in the text the reference of the study we mention.

Line 268 ABR has not been defined

ABR meaning is defined in line 194.

Line 280 – that vs than

The mistake kindly noticed by the reviewer is now corrected.  

Line 287 The hypothalamus

Thank you for your observation.

Line 301 thermoregulation vs thermoregulatory

This mistake is now corrected. Thank you.

Line 320 – how important is this reduction in appetite though in terms of HS?

Data available to date with either metformin or other drugs prevent knowing how important the effect of appetite reduction in HS is. The aim of our review was to compile drugs and excipients that may affect hydration status owing to their mechanism of action or adverse effect.

Line 333 grammar

Following the reviewer´s suggestion, we have modified the text.  

Line 345 being controversial. ‘Additional’ rather than ‘extraordinary’.

The proposed change has been included in the new version of the manuscript.

Line 356 – citation needed – which study?

As requested by the reviewer, we have included the citation needed.

Line 361 citation needed

The requested citation is now included in the manuscript.

Line 371 What are the other contributors to AAD according to this/other studies

We have modified the text for a better understanding of this issue.

Line 373 Links

This grammatical error is already corrected in the text.

Line 378 Increased fatality rates

Reviewer´s suggestion is now included in the manuscript.

Line 442 Occurs

We have changed the verb attending to reviewer´s proposal.  

Line 467 ‘effects occur’

This grammatical mistake has been corrected.

Line 470 ‘is crucial to’. What are the complications referred to here?

Hypohydration consequences are mentioned in the Introduction of this manuscript (Lines 89-91). Thus, we consider it is no necessary to indicate them again in this section.

Line 474 – how prevalent is NSAIDs use?

In the new version of the manuscript data of defined daily doses (DDD) by the WHO are included.

Line 492 chronically

Thank you for your observation.

Line 509 – Was this chronic or acute diarrhoea? You mention the importance of the distinction in the introduction, but then do not really address this issue in the remainder of the review.

Thank you for your comment. We have added a new statement to clarify this issue. Moreover, in the introduction section, as previously stated, we have included the differences between acute and chronic diarrhoea (acute diarrhoea lasts less than four weeks and have usually infectious origin whereas chronic diarrhoeas are classified into watery, inflammatory or fatty diarrhoeas, based on stool characteristics).

Line 558 Specify these consequences

As previously stated, hypohydration consequences are indicated in the Introduction of this manuscript (Lines 89-91). Hence, we consider it is no necessary to mention them again.

Line 564 SIADH

We have corrected the detected mistake in this acronym.

Line 607 – what is a ‘high incidence’?

Thank you very much for your suggestion. Lithium salts relationship with polyuria, diarrhoea, enuresis and polydipsia was compared to that of other antipsychotic drugs such as valproate.

Line 612 ‘since it relaxes’ ‘These drugs’

Thank you for your observation in these points.

Line 617 ‘Raises’ …..’similarly to’ not ‘with’

This grammatical error has been corrected in the new version of the manuscript.

Line 631 ‘highlighted that’

This mistake has been corrected.

Line 652-3 grammatical issues

According to the reviewer´s suggestion, we have reviewed this sentence to correct grammatical errors.

Line 654 – sentence is a bit confusing – rewording required e.g. ‘contemplate’

This sentence was rewritten following the reviewer´s suggestion.

Line 659 coordinated

This sentence was rewritten in the new version of the manuscript.

Overall response

This is an interesting review of a little explored topic. It could be improved by identifying the limitations of the review e.g. narrative review as opposed to a systematic review. Also the quantification of the impact of these e.g. on balance for each drug/drug class - are these effects minor, moderate, major magnitude? Similarly, the prevalence of these side effects is only occasionally addressed - could be more consistently addressed where the data supports.

We would like to acknowledge reviewer´s comments for the improvement of this review. We have included in the new version of the manuscript a Strengths and Limitations section.

Round  2

Reviewer 1 Report

The manuscript has been modified based on most of the previous recommendations, so well done.

It however will still benefit from revision by a linguistic for grammar and fluency.

This will make the content more accessible to your readers.

I hope this can be done as soon as possible.

Author Response

The manuscript has been modified based on most of the previous recommendations, so well done.

Thank you very much.

It however will still benefit from revision by a linguistic for grammar and fluency. This will make the content more accessible to your readers. I hope this can be done as soon as possible.

According to the reviewer`s suggestion, the text has undergone English language editing by MDPI (Certificate attached).

Reviewer 3 Report

Happy to approve for publication if the following corrections not rectified to my satisfaction are done:

Line 85 (latest draft) ‘Alteration of the’ should read ‘Altered renal function and……’

ABR

Line 268 ABR has not been defined

ABR meaning is defined in line 194.

It should be ARBs rather than ABR

 Line 474 – how prevalent is NSAIDs use?

In the new version of the manuscript data of defined daily doses (DDD) by the WHO are included.

Prevalence differs from typical dose – prevalence denotes the % of the population taking the drug in a period of time

Author Response

Happy to approve for publication if the following corrections not rectified to my satisfaction are done:

We would like to thank reviewer´s suggestions to improve our manuscript.

Line 85 (latest draft) ‘Alteration of the’ should read ‘Altered renal function and……’

This mistake kindly noticed by the reviewer has been corrected in the new version of the manuscript (line 144).

ABR

Line 268 ABR has not been defined

ABR meaning is defined in line 194.

It should be ARBs rather than ABR

This mistake is already corrected. Thank you.

Line 474 – how prevalent is NSAIDs use?

In the new version of the manuscript data of defined daily doses (DDD) by the WHO are included.

Prevalence differs from typical dose – prevalence denotes the % of the population taking the drug in a period of time

According to the reviewer´s suggestion we have included prevalence data in the new version of the manuscript (lines 1326-1331).

Finally, we would like to inform the reviewer that the manuscript has undergone English language editing by MDPI
